# Population Parameters of *Haplaxius crudus* (Hemiptera: Cixiidae) under Semi-Controlled Conditions

**DOI:** 10.3390/insects15020085

**Published:** 2024-01-26

**Authors:** Ivette Johana Beltrán-Aldana, Anamaria Fernández-Sánchez, Anuar Morales-Rodriguez

**Affiliations:** 1Pest and Disease Program—Entomology Area, Colombian Oil Palm Research Centre (CENIPALMA), Centro Empresarial Pontevedra, Calle 98 No. 70-91, Piso 14, Bogotá D.C. 111211, Colombia; amorales@cenipalma.org; 2Agronomy Engineering Program, Universidad de Cundinamarca, Facatativá 253058, Colombia; anamariaanamaria9@gmail.com

**Keywords:** Colombia, life table, palm leafhopper, lethal wilt disease, *Elaeis guineensis*

## Abstract

**Simple Summary:**

Oil palm is one of Colombia’s main agribusinesses, with more than 590,000 ha planted. The eastern palm-growing region accounts for 47% of the planted area. It is also the area affected by the Lethal Wilt disease, which has caused the eradication of more than a million palms, equivalent to 7790 ha. *Haplaxius crudus* is considered a vector of the causal agent of this disease, and management strategies are focused on reducing *H. crudus* populations in the field. However, the population parameters of the insect, which are essential for adequate control, are unknown. Therefore, it was proposed to determine the life cycle and life table parameters under semi-controlled conditions in an oil palm plantation. The life cycle was 62.7 ± 15.5 days (average ± standard deviation). The egg stage lasted 14.6 ± 0.6 days, the nymphal stage 48.1 ± 14.9, and adult longevity was 14.8 ± 8.4 days. The highest specific mortality rate (qx) was in the egg phase: 0.14. The net reproductive rate was Ro = 10.96; the generation time was T = 62.3 days, the intrinsic natural growth rate was rm = 0.03, the and finite growth rate was λ = 1.03. These results contribute to the knowledge of the population dynamics of this insect in the field and for the development of population control studies.

**Abstract:**

The palm leafhopper, *Haplaxius crudus*, is a possible vector of the pathogen that causes the Lethal Wilt of oil palms in Colombia. This disease represents the biggest phytosanitary problem in the eastern palm zone. From 2010 to 2021, more than 7700 ha have been eradicated, with economic losses exceeding 154 million USD. Therefore, knowing the biology of this insect and its population parameters is necessary for developing population control tools. To evaluate these parameters, a cohort of 100 eggs obtained from *H. crudus* adults from the breeding unit established in the Campo Experimental Palmar de las Corocoras de Cenipalma in Paratebueno, Cundinamarca, was monitored to record the life cycle and the population parameters using a life table under semi-controlled conditions in an oil palm plantation. The life cycle from egg to adult was 62.7 ± 15.5 days (26.1 ± 2.9 °C; HR: 89.8 ± 14.0%). The egg stage lasted 14.6 ± 0.6 days, the nymphal stage 48.1 ± 2.8 days through five instars, and the adult longevity was 14.8 ± 8.4 days. The specific mortality rate (qx) calculated in the life table was 0.14 (for the egg stage), 0.05 (for I instar), 0.05 (for II instar), 0.03 (for III instar), 0.04 (for IV instar), and 0.07 (for V instar). The population parameters’ values were as follows: net reproductive rate Ro = 10.96; generation time T = 62.3 days, intrinsic natural growth rate rm = 0.03, and finite growth rate λ = 1.03. These results help us to understand the population dynamics of this insect in the field and for the development of population control studies.

## 1. Introduction

*Haplaxius crudus* Van Duzee was described in Jamaica in 1907 and is native to southern Florida, Cuba, the Cayman Islands, Jamaica, Trinidad, and tropical America, from Mexico to northern South America [1,2]. The adult *H. crudus* is a vector of the causative agent of lethal coconut yellowing (ALC), a highly destructive disease of coconut and ornamental palms in Florida and several countries in the Caribbean Basin region [1,3]. The lethal yellowing of the coconut palm is caused by the phytoplasma *Candidatus* Phytoplasma palmae, belonging to the group 16SrIV subgroup A [4,5]. Studies conducted by Cenipalma found that *H. crudus* is responsible for disseminating in palm plantations in Colombia, the pathogen causing Lethal Wilt (LW). This disease represents the most significant phytosanitary problem for oil palm in the Colombian eastern palm zone. The first cases of LW were recorded in Bajo Upía (Casanare, Colombia) in 1994 [6]. According to data from Cenipalma, between 2010 and 2021, there were 1,114,025 palms that were eradicated due to LW [7] which has generated economic losses of more than 154 million USD [8].

Palms affected by the LW show rotting of the inflorescences and fruit bunches. In addition, a dark brown drying occurs on the leaves that begins from the apex of the leaflets towards the base. These symptoms are initially observed in one or two middle-third leaves and then they spread to all the leaves without defined uniformity [9,10].

*Haplaxius crudus* undergoes incomplete metamorphosis, i.e., egg, nymph, and adult stages, and it completes its entire biological cycle in grasses and Cyperaceae and the adults feed on oil palms. The egg is placed by the females in the grasses growing around the palms and in the plantation paddocks; the eggs are inserted near the base of the stem and below the leaf sheaths that begin to dry out [10,11]; when the nymphs emerge, they are located on the roots of the grasses to feed and they remain in the nests that they produce, with a kind of silk that they secrete from their abdominal glands [3]. When they reach the adult stage, they fly to the foliage of the palm, and in the underside of the leaves, they mate and feed by sucking phloem fluids [1,10]. In this way, feeding on diseased palms and then on healthy palms, could transmit the pathogen that causes the disease. The symptoms of this disease in oil palms appear approximately six months after being infected [9,10,12].

There are some reports of natural enemies of *H. crudus* present in Mexico and Florida (USA), including spiders of the genus *Theridion* (Walckenaer) (Araneae: Theridiidae) and the ant *Solenopsis invicta* (Buren) (Hymenoptera: Formicidae) [13,14], parasitic mites of the genera *Leptus* (Latreille) (Trombidiformes: Erythraeidae) and *Erythraeus* (Latreille) (Trombidiformes: Erythraeidae) [14] and the fungus *Hirsutella citriformis* (Speare) (Hypocreales: Ophiocordyetaceae) [14,15]. In Colombia, it was established that a strain of the fungus *Metarhizium anisopliae* (Metchnikoff) Sorokin (Hypocreales: Clavicipitaceae) affects adults and nymphs of *H. crudus*. However, the most widespread control measure for this insect in plantations is the calendar application of systemic chemical insecticides (neonicotinoids), which incurs higher production costs and affects beneficial fauna in the plantations [16].

Although several investigations have been carried out that focus on the control of *H. crudus* [15,16,17], few studies are related to the biology of the insect [18,19], such as its population parameters. Information that allows an estimate of what is happening in nature and knowledge of the biological attributes of the pest, offers essential information for its understanding and management [20]. The objective of this research was to study the biology of *H. crudus* on the grass *Panicum maximum* L. cv Mombaza, host of eggs and nymphs of the insect, and its relationship with the palm oil, *Elaeis guineensis*, host to its adults. The study was carried out under semi-controlled conditions in a shade house in an oil palm plantation to determine the development time and morphological characteristics of each stage and to estimate age-specific mortality parameters, reproductive capacity, and adult longevity.

## 2. Materials and Methods

### 2.1. Location

The research was conducted in the *H. crudus* breeding unit in the Experimental Field “Palmar de Las Corocoras” (CEPC) of Cenipalma in Paratebueno, Cundinamarca, at an altitude of 227 m a.s.l., latitude 4°22′04′′ N and longitude 73°10′16′′ W. The average temperature during the study was 26.1 ± 2.9 °C and the relative humidity was 89.8 ± 14% (WatchDog^®^ Logger 3617A, Aurora, IL, USA), and the natural day length was 12 h.

### 2.2. Egg Cohort

A cohort of 100 eggs obtained from one day of oviposition of 50 eight-day-old gravid females of *H. crudus* from the breeding unit was used. These females were released inside oviposition units (Figure 1A) made up of acetate cylinders (35 cm high and 7 cm in diameter), modified with two side windows (5 cm × 8 cm), and covered with tulle fabric to allow for the circulation of air. At the upper end, they were sealed with a foam (2.5 cm thick) to which a cut was made in the center to introduce the oil palm leaflets for feeding the *H. crudus* females, and at its base, there was a small pot (7 cm diameter) with a clump of *Digitaria sanguinalis* L. (Poales: Poaceae) for egg laying, which was sown using coconut fiber-based plugs as the substrate (Figure 1B). The *D. sanguinalis* clumps infested with eggs (Figure 1C) were examined under a stereoscope and with the help of a dissection needle and a fine brush, the eggs were removed and deposited in Petri dishes (5.5 cm diameter) conditioned as a humid chamber for incubation.

### 2.3. Life Cycle

The eggs were observed daily to record their characteristics and development time until hatching. When the nymphs emerged, they were immediately isolated in cylindrical pots (4 cm diameter × 5 cm high) with a clump of *P. maximum*, with a good root system planted in coconut fiber plugs, to monitor each instar’s characteristics and development time. When the adults emerged, they were placed in pairs (one ♀ and one ♂) within oviposition units to record the longevity of sex, the oviposition potential of the females, and the hatching rate of their eggs. When a male died, it was replaced by another one obtained from the breeding unit.

### 2.4. Population Parameters

With the daily record of the number of survivors in each of the development phases of the insect, the life table was established based on the complete cohort method or horizontal, in which a group of individuals of the same age is followed from birth throughout their lives [21,22]. The parameters used in the life table were: age (x) which designates the exact age of each individual *H. crudus* from birth to death; number of survivors at age x (nx); fraction of the original cohort that dies in the interval between age x and the following (dx), fraction of the cohort that survives between two successive instars (qx), the proportion of individuals surviving at age x (lx) and net fecundity (mx) that represents the number of females obtained for each female of age x. This information allowed for the estimation of the net reproductive rate, which is defined as the average number of females produced by each female per generation:Ro = Σ(mxlx)

The generation time of the cohort, defined as the average age of females in a cohort at the time of the birth of its offspring is as follows:T = Σxlxmx/Ro

The intrinsic rate of natural growth of the population, which expresses the capacity of the increase in the number of females per female per unit of time is as follows:rm = ln R0/T

And, the rate of finite increase, that indicates the number of individuals added to the population per female and per unit of time [22,23] is as follows:λ = erm

## 3. Results

### 3.1. Description of the Different Stages

Eggs. They are inserted individually or in groups (two to ten) in the senescent leaf sheaths. They are fusiform and smooth, pointed at the apical end and rounded at the basal end. Initially, they are translucent with white ends and when close to hatching, they turn whitish and have a curved appearance. From the fifth day, the eyes can be distinguished, which present on each side as two small red spots (Figure 2A).

Nymphs. The nymphs have five instars; their color varies between light yellow and beige, and their legs are white and long in symmetry with the body. Waxy exudate formation is seen in segments 7, 8, and 9 of the abdomen. With this exudate, they form nests to protect themselves from the environmental conditions and some natural enemies [11]. The most significant characteristic to determine the instar, in which the nymph is found, is the length of the wing primordia, which grows with each molt until they reach the posterior edge of the thorax.

Instar I. The newly emerged nymphs are white, and later the head and abdomen turn light yellow. The eyes are red, and the antennae protrude beyond the frame of the head; the most outstanding characteristic of this instar is that the thorax is larger than the abdomen (Figure 2B).

Instar II. The nymph is beige on the thorax and light yellow on the abdomen. The size of the abdomen is like the size of the thorax, and its segmentation is more marked (Figure 2C).

Instar III. The size of the nymphs in this instar increases notably relative to the previous instars. The abdomen and thorax continue with the same coloration as the previous instar. When the nymph is close to molting at instar IV, the primordia of the forewings become more noticeable, which are attached to the mesothorax and reach a third of the metathorax in this instar (Figure 2D).

Instar IV. The nymphs in this instar begin to widen their entire body (thorax and abdomen) and as they recently emerged, they present a darker coloration; the primordia of the forewings are more developed, reaching two-thirds of the metathorax (Figure 2E).

Instar V. In this instar, the nymphs increase in size significantly; as they recently emerged, they are light brown, and later, the thorax turns light yellow and the abdomen greenish yellow. The anterior wing primordia are longer, protruding from the metathorax and reaching the third segment of the abdomen. The eyes increase in size, and their red coloration becomes intense. The waxy filaments are longer and more abundant compared to the other instars (Figure 2F).

Adult. The recently emerged adult takes a few hours to start flying for food (Figure 2G). Its main characteristic is the burgundy compound eyes, which turn light yellow 24 h later. The body is light yellow, with a slightly greener abdomen, short setiform antennae, and transparent membranous wings. The differentiation between female and male is apparent from the genitalia of the female, in which an ovipositor is observed in the ventral area of the abdomen. In a preliminary stage of this study and using the same methodology, it was observed that the 8-day-old females could lay an average of 69.3 ± 23.3 eggs and a maximum of 124 eggs, and the 15-day-old females could lay an average of 85.8 ± 22.7 eggs with a maximum of 137 eggs.

### 3.2. Lifecycle

The egg development period was 14.6 ± 0.6 days, and the nymphal stage was 48.1 ± 14.9 days, passing through five instars. Adult longevity was on average 18.2 ± 8.7 days for females and 11.2 ± 9.3 days for males (Table 1).

### 3.3. Population Parameters

The highest mortality rate (qx) occurred in the egg stage (qx: 0.14), followed by the V nymphal instar (qx: 0.07), while the instar with the lowest mortality rate was instar III (qx: 0.03) (Table 2). Likewise, in population studies, the value of Ro indicates that 10.96 females replaced each female in one generation and that the population was growing (Ro > 1). The capacity for increase or intrinsic rate of population growth had a value rm = 0.03; each female can contribute 0.03 females/day throughout the generation time, and the finite growth rate was λ = 1.03, indicating that 1.03 individuals/day are added to the population. The generation time (T) of the cohort was 62.3 days. That is to say that the females at the time of the birth of their first offspring were of that average age. All this information would mean that approximately six generations per year could be obtained under the conditions prevalent during the study.

## 4. Discussion

Although the literature related to the biology of this insect is little, studies carried out by Tsai and Kirsch [18] found an egg stage duration of 11 ± 0.0 days and a nymphal stage of 41.6 ± 12.0 days at a temperature of 30 °C and a duration of 19.5 ± 0.8 and 61.3 ± 13.3 days, respectively, for eggs and nymphs at 24 °C. In this study, adult longevity was 7.8 days for females and 7.3 days for males at 24 °C. Other laboratory studies indicate that the egg stage lasted 12–13 days (T: 24–28 °C), and adult emergence began 7–9 weeks later the emergence of the nymphs [24]. Piña [25], during samplings in the urban area of Mérida, Yucatán (mean annual temperature between 25 and 26 °C), reported that each of the five nymphal stages lasted 9 days. Halbert et al. [26] mention that the number of generations of this insect is affected by temperature. However, even this effect of temperature allows us to observe different growth rates of individuals within the same niche [18]. Velasco and Walter [27] reported that insect survival, nymphal growth, and reproductive rate were highly influenced by food quality. Effects due to host variations (rootstocks and orange scions) were also observed in *Diaphorina citri* Kuwayama, 1908 (Hemiptera: Liviidae) [28,29], and due to environmental conditions such as temperature, as evidenced in *Sipha flava* Forbes, 1884 (Hemiptera: Aphididae), fed with elephant grass [30].

In the literature there is no information on the reproductive parameters of *H. crudus*; however, Table 3 allows comparison of the population parameters of *H. crudus* and other leafhopper vectors of pathogens such as *Nilaparvata lugens* (Stål, 1854) (Hemiptera: Delphacidae), a rice pest in Asia and Oceania [31]; *Peregrinus maidis* (Ashmead, 1890) (Hemiptera: Delphacidae), a pest of maize and sorghum and vector of the “Maize Mosaic Rhabdovirus” and “Maize Tenuivirus” viruses in tropical environments [32]; *Sogatella furcifera* (Horváth, 1899) (Hemiptera: Delphacidae), a pest of rice in South Asia [33]; *Lacertinella australis* (Remes Lenicov & Rossi Batiz, 2011) (Hemiptera: Delphacidae), a pest of grasses of the genus *Cortaderia* spp. and present in garlic, rice and rye crops in Argentina [34]; and *D. citri* vector of the bacterium that causes the disease called “huanglongbing” (HLB) in citrus [28]. Compared to the other leafhoppers, *H. crudus* presents one of the highest net reproductive rate (Ro) values, like those recorded for *N. lugens*. Regarding the intrinsic growth rate (r), the lowest value corresponds to *H. crudus*, similar to that obtained for *L. australis*; however, a value greater than zero indicates that the population is growing. Likewise, *H. crudus* presents the highest estimated value for generation time (T). This indicates the development of a few generations per year; however, its role as a vector of a lethal disease in oil palm is still worrisome.

The r value helps compare the growth potential of a population with other species. In this case, it is lower in *H. crudus* than in the other four species. Nevertheless, its positive value indicates the importance to this insect which serves as a possible vector of the pathogen that causes LW since a single specimen that survives management strategies is a potential carrier of the disease [35].

In conclusion, a high net reproductive index (Ro), a positive intrinsic growth rate (r), and a short generation time makes *H. crudus* a pest that can rapidly increase its population size, with the aggravating factor that it is an insect that serves as a vector of the pathogen that causes lethal wilt, a devastating disease for oil palm in Colombia. Therefore, it is necessary to focus our attention on the effective combination of management practices that allow the reduction of *H. crudus* populations and, as a direct effect, reduce the speed of the spread of this disease.

## Figures and Tables

**Figure 1 insects-15-00085-f001:**
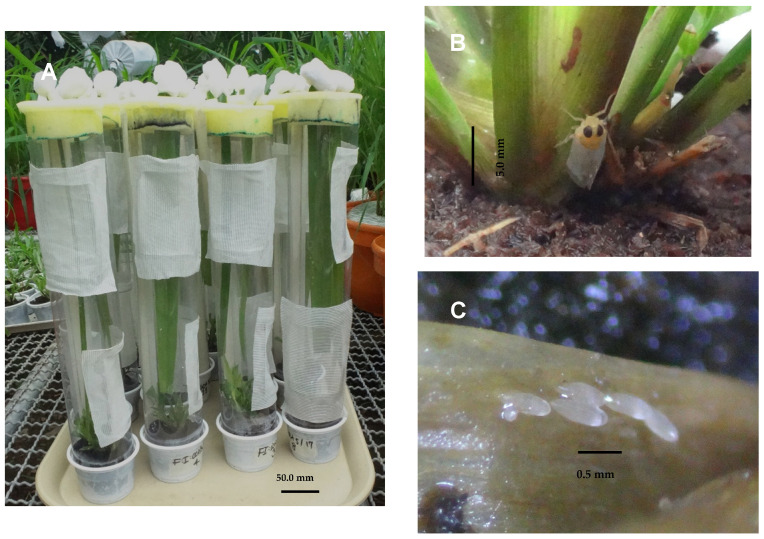
Methodology for obtaining *Haplaxius crudus* eggs. (**A**) Oviposition units; (**B**) ovipositing female; and (**C**) eggs inserted in the leaf sheath.

**Figure 2 insects-15-00085-f002:**
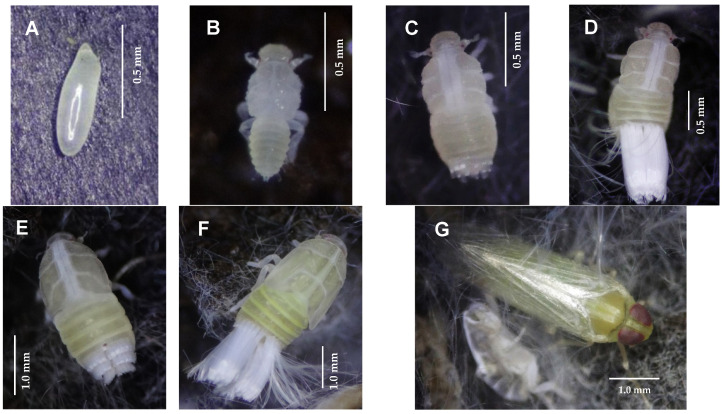
Developmental stages of *Haplaxius crudus*. (**A**) Egg; (**B**) nymph I; (**C**) nymph II; (**D**) nymph III; (**E**) nymph IV; (**F**) nymph V; and (**G**) teneral adult.

**Table 1 insects-15-00085-t001:** *Haplaxius crudus* life cycle (under shade conditions: T: 26.1 ± 2.91 °C; HR: 89.8 ± 14%).

Stage	Number	Average ± SD * (Days)	Range (Min-Max)
EGGS	100		
Oviposition to hatch	86	14.6 ± 0.6	11–20
NYMPHS			
I instar	86	9.2 ± 2.3	7–12
II instar	82	7.5 ± 3.0	6–11
III instar	78	7.0 ± 1.5	4–12
IV instar	76	9.9 ± 2.1	7–15
V instar	73	14.2 ± 3.2	7–23
Total nymphs		48.1 ± 14.9	
ADULTS	68		
Females	33	18.2 ± 8.7	8–34
Males	35	11.2 ± 9.3	5–27
Average life span	68	14.8 ± 8.4	

* SD = standard deviation.

**Table 2 insects-15-00085-t002:** *Haplaxius crudus* life table (under shade conditions: T: 26.1 ± 2.91 °C; HR: 89.8 ± 14%).

Stage (*X*)	*n_x_*	*d_x_*	*q_x_*	*l_x_*	*m_x_*	*R_o_*
Egg	100	14	0.14	1.00	-	-
I instar	86	4	0.05	0.86	-	-
II instar	82	4	0.05	0.82	-	-
III instar	78	2	0.03	0.78	-	-
IV instar	76	3	0.04	0.76	-	-
V instar	73	5	0.07	0.73	-	-
Adult	68	68	1.00	0.68	16.12	10.96

*n_x_*: number of individuals that survive to the beginning of the age interval *x*; *d_x_*: number of individuals that die between ages *x* and *x* + 1; *q_x_*: the probability of dying between *x* and *x* + 1; *l_x_*: proportion of survival at age *x*; and *m_x_*: average number of female progenies produced by each female of age *x*. *R_o_*: net reproductive rate.

**Table 3 insects-15-00085-t003:** Comparison of the population parameters of *H. crudus* with other insect vectors of phytopathogens.

Species	Parameters				References
	Net Reproductive Rate (*R_o_*)	Intrinsic Growth Rate (*r*)	Rate of Finite Increase (*λ*)	Generation Time (*T*)	
*Haplaxius crudus*	10.96	0.038	1.039	62.33	
*Nilaparvata lugens*	10.02	0.067	1.068	34.05	[31]
*Sogatella furcifera*	9.27	0.069	1.080	31.86	[33]
*Peregrinus maidis*	7.34	0.044	-	44.90	[32]
*Lacertinella australis*	12.75	0.037	1.037	64.41	[34]
*Diaphorina citri*	8.078	0.087	-	23.96	[28]

## Data Availability

Data supporting this research are available from the corresponding authors upon request.

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
