# Peer review of "Population Parameters of Haplaxius crudus (Hemiptera: Cixiidae) under Semi-Controlled Conditions"

_insects, 2024, doi:10.3390/insects15020085_

Round 1

Reviewer 1 Report

Comments and Suggestions for Authors

Population parameters of Haplaxius crudus (Hemiptera:Cixiidae) under semi-controlled conditions

Major comments

Figures that are presented do not have any scale bars.  Without scale bars, the size of the plants used and eggs are unknown (Figure 1) and the growth of the insect (Figure 2) is unclear.  All figures should have scale bars added to aid the reader in the interpretations that are indicated in the text.

In the discussion, the fact that r is positive is stressed, but how does the rate of increase vary in the field rather than in the semi-controlled condition?  What is the rate of disease spread that is meadured in the region where the work occurred?  What is the rate of increase of the biological control systems that may aid in reducing the r in the field?  Those question need to be addressed before warning about the problem with the insect spreading diseases.

Minor comments

Page 1

Line 37.  “it is believed” is a weak beginning of a sentence.  Do you mean “H. crudus is responsible …” if not say work is still necessary to determine how disease is spread.

Line 39.  Add “Colombia” after “(Casanare)” as all readers are not familiar with the locale.

Line 44. Clarify what “bunches” are.  Are these the immature fruit?

Page 2

Line 55. Is “sap” phloem or xylem fluids?  Be specific regarding feeding region of insect.

Line 67-8.  Are the insecticides “chemically synthesised” or do you mean something else?  Is there a specific insecticide (e.g. insect growth regulators) that are used?

Page 3

Line 119. Change “individualized” to “isolated”.

Line 120. Italicize “P. maximum”

Line 122. Change “individualized” to “placed”

Line 127-8.  What does “horizontal” used here mean?

Separate equations from rest of text to highlight equations.

Page 4

Line 144. Give range of eggs inserted into leaf sheaths, not just “groups or individually”

Line 151-2.  Role of exudate is discussion.  How did you determine that this protects the insect from humidity or that it has any role in changing water loss pattern of insect?

Page 5

Line 189. Define “serosity” and explain how it increases in last instar rather than “production is higher”

Line 191. Insert space between full stop and 2 “(Fig.2G)”

Line 203. Delete “(T:26.1ËšC±2.9ËšC)”

Page 7.

Line 253. Delete “While” at beginning of sentence.

Line 261.  Unclear reason for citing work on beetle here unless the work was done near where your work is done.

Line 276. Italicize “N. lugens”

Page 8

Lines 287-291.  Re-write paragraph to explicitly state that disease could spread faster as a result of a single infected female with the high reproductive rate assuming the disease is passed vertically.

Comments on the Quality of English Language

English is acceptable

Author Response

Dear Reviewer, 

We want to thank you for the comments and suggestions to our manuscript, we are sure all of them improve our work. 

Reviewer 2 Report

Comments and Suggestions for Authors

Line 18: a life table

32,36: In 36 is repeated what is stated in 31-32 with a new reference. Do not repeat.

37, Write: It is also believed.... for disseminating the pathogen  causing...

45, begins

49, goes through incomplete metamorphosis, i.e. egg,........

50, what grasses? Give more deail (growing under the palms?)

51, on or in the roots?

52, nests is a srange concept here, normally used for birds. Use "nests" or a different word, such as enclosures or structures.

60, italics for Theridion

64, reference for the fungus?

68, delete synthesis

70, that focused

72,  use "such as"' for "leaving aside"

74, write: are essential information for its understanding and management

120, italics for P. maximum

146, delete ;

149, write: the nymphs have five .....colour

152, "nests".

152, I have never heard of an insect protecting itself against humidity. Perhaps rain drops. You need a reference or be more speculative  and write: possibly to protect itself....

156, perhaps: antennae protrude beyond the frame of the head

177, write: notably relative to the ..

180, born in? Perhaps attached to

194, write: male is apparent from..

232, You did not evaluate the conditions. Write: prevalent during the study.

253, delete while

264, information on the

265, comparison with other

276, N. lugens in italics

280, lower number of generations than the other species? Furthermore, its role as a vector is not relevnt here. Delete.

287, compare with other species. Or: estimate the growth potential

288, But you have 5 other species, 4 with higher values 

288, Write: positive value indicates the importance of his insect which serves as

293, population

Comments on the Quality of English Language

See above, I made several suggestions for improvement.

Author Response

(The authors gave the same response as above.)
